



# Assessment of Real-time Bioaerosol Particle Counters using Reference Chamber Experiments

Gian Lieberherr[1], Kevin Auderset[2], Bertrand Calpini[1], Bernard Clot[1], Benoît Crouzy[1],
Martin Gysel-Beer[3], Thomas Konzelmann[1], José Manzano[4,6], Andrea Mihajlovic[5], Alireza Moallemi[3],
David O'Connor[4,8], Branko Sikoparija[5], Eric Sauvageat[1,7], Fiona Tummon[1], and Konstantina Vasilatou[2]

[1]Federal Office of Meteorology and Climatology MeteoSwiss, Payerne, Switzerland
[2]Federal Institute of Metrology METAS, Bern-Wabern, Switzerland
[3]Laboratory of Atmospheric Chemistry, Paul Scherrer Institute, Villigen, Switzerland
[4]Technological University of Dublin, Dublin, Ireland
[5]BioSense Institute - Research Institute for Information Technologies in Biosystems, University of Novi Sad, Novi Sad, Serbia
[6]Now at the Technical University of Munich, Munich, Germany
[7]Now at the Institute of Applied Physics and Oeschger Centre for Climate Change Research, University of Bern, Bern,
Switzerland
[8]Dublin City University, Dublin, Ireland

**Correspondence:** Konstantina Vasilatou (Konstantina.Vasilatou@metas.ch), Gian Lieberherr
(gian.lieberherr@meteoswiss.ch)

**Abstract.** This study presents the first reference calibrations of three commercially available bioaerosol detectors. The Droplet Measurement Technologies WIBS-NEO, Plair Rapid-E, and Swisens Poleno were compared with a primary standard for particle number concentrations at the Federal Institute for Metrology METAS. Polystyrene (PSL) spheres were used to assess absolute particle counts for diameters from 0.5 μm to 10 μm. For the three devices, counting efficiency was found to be

strongly dependent on particle size. The results confirm the expected detection range for which the instruments were designed. While the WIBS-NEO achieves its highest efficiency at smaller particles, e.g. 90% for 0.9 μm diameter, the Plair Rapid-E performs best for larger particles, with an efficiency of 58% for particles with a diameter of 10 μm. The Swisens Poleno is also designed for larger particles, but operates well from 2 μm. However, the exact counting efficiency of the Poleno could not be evaluated as the cut-off diameter range of the integrated concentrator unit was not completely covered. In further experi-

ments, three different types of fluorescent particles were tested to investigate the fluorescent detection capabilities of the Plair Rapid-E and the Swisens Poleno. Both instruments showed good agreement with the reference data. While the challenge to produce known concentrations of larger particles above 10 μm or even fresh pollen particles remain, the approach presented in this paper provides a potential standardised validation method that can be used to assess counting efficiency and fluorescence measurements of automatic bioaerosol monitoring devices.


## 1   Introduction

Routine pollen monitoring is carried out across the world, particularly in developed countries where the prevalence of allergies is highest (D'Amato et al., 2007; Wüthrich et al., 1995; Ring et al., 2001; Buters et al., 2018). At present, monitoring networks rely nearly exclusively on manual instruments developed in the 1950s (Hirst, 1952). These samplers continuously collect
airborne particles on a rotating drum, which is removed from the instrument on a regular schedule, usually once per week. Thereafter the tape is disassembled from the drum and pollen manually identified and counted using optical microscopy (Galán et al., 2014). The time-consuming nature and professional expertise required for this method restricts both the possible number of stations across a network and the time-resolution of data provided to end users (usually daily averages).

Several automatic bioaerosol monitors have come to market over the past few years, allowing real-time or near real-time
observations at high temporal resolution (for a recent overview see (Huffman et al., 2019)). These include the Droplet Measurement Technologies WIBS, which is a three-channel spectrometer (Könemann et al., 2019; Perring et al., 2015a; Foot et al., 2008) that has been widely used for a range of research purposes. Initial studies focused on just the differentiation between fluorescent and non-fluorescent particles (Gabey et al., 2010), while more recently a typing scheme has been applied to provide information about aerosol fluorescent properties which are linked to different particle types, including bacteria, fungal
spores, and pollen (Savage et al., 2017; Hernandez et al., 2016; Perring et al., 2015a; Ruske et al., 2018). Results, however, varied between different instrument versions and no classification between different pollen taxa is currently possible. Two more recently-developed instruments also based on airflow cytometry, the Plair Rapid-E (Šauliene et al., 2019; Kiselev et al., 2013; Crouzy et al., 2016) and Swisens Poleno (Sauvageat et al., 2020), have shown the potential to classify pollen taxa. This is an essential requirement for pollen monitoring since allergy-sufferers, one of the main end-user groups of the information these
networks provide, are sensitive to specific pollen taxa above particular concentration thresholds (Jaeger, 2006).

As interest grows in using automatic instruments to provide real-time information about airborne pollen concentrations, it is vital that standard calibration methods are developed that can be applied across monitoring networks. Such techniques do not currently exist for automatic pollen monitoring, in part because of the novelty of real-time instruments and also because no method currently exists to aerosolise a known quantity of a particular pollen taxon. This in turn is a result of the fact that
pollen particles are considerably larger than the aerosols that are typically monitored in air quality networks and because they are very fragile, being easily broken up into smaller pieces using traditional aerosolising methods. Since the above-mentioned monitoring devices rely on light-induced fluorescence to identify single particles, a validation of the fluorescence response to a known excitation wavelength is also a crucial step in instrument validation.

In this study we present results from the first reference calibrations of three real-time bioaerosol monitors, the Droplet
Measurement Technologies WIBS-NEO, Plair Rapid-E, and Swisens Poleno, using primary standards at the Federal Institute for Metrology METAS. Section 2 outlines the experimental set-up and instruments used. Section 3 presents the results and a discussion of the performance of each instrument in terms of absolute number counts, counting efficiency, uncertainty and stability, particle size detection as well as the fluorescence measurements. The conclusions are provided in Section 4.



## 2 Experimental Setup and Instruments Tested

### 2.1 Primary standard for particle number concentration at METAS

The primary standard for particle number concentration at METAS is a large-scale facility designed for calibration of optical and aerodynamic particle size spectrometers (OPSSs and APSSs, respectively). The design has been described in detail elsewhere (Horender et al., 2019), and only the main aspects are briefly presented here. The experimental setup consists of three distinct sections: the aerosol generation system, a turbulent-flow tube (aerosol homogeniser), and a reference optical particle

counter (Horender et al., 2019). Polystyrene (PSL) spheres in the size range 100 nm - 10 μm are aerosolised using methods based on wet or dry dispersion with commercial and custom-made generators. It is known that PSL suspensions contain additives, such as surfactants to prevent particles from agglomerating. Upon aerosolisation, these additives give rise to so-called residue particles which can create unwanted peaks in the particle distribution. In such cases, the PSL particles are selected according to size using a differential mobility analyser (DMA; TSI Inc., USA) or an aerodynamic aerosol classifier (AAC;

Cambustion Ltd., UK) to filter out unwanted residue particles.

The aerosol homogeniser is a 4 m-long custom-made stainless-steel tube with an inner diameter of 16.4 cm, placed vertically to prevent deposition losses of large particles. Dry filtered air enters the homogeniser from the top (at a fixed flow rate of 120 L/min) and sweeps the PSL particles down the tube. The particles are subsequently mixed by three turbulent air-jets. The air-jet injection tubes are placed symmetrically around the homogeniser tube pointing downwards at an angle of 60 °. The

flow profile has been characterised using Laser Doppler Velocimetry (LDV) and was found to be turbulent (plug flow, with negligible effect of the laminar sub-layer). The sampling zone is located 3.0 m downstream of the aerosol injection position and accommodates two isokinetic sample probes, one for the device under test and one for the reference optical particle counter. The aerosol homogeneity at the different sampling ports has been determined experimentally to be within 1.1% in number concentration (95% confidence level) (Horender et al., 2019).

The reference detection system consists of a custom-made optical particle counter placed at the outlet of the aerosol homogeniser (Horender et al., 2019). Connection tubes are kept as straight as possible to avoid impaction losses of large particles. The sampled aerosol enters the detection chamber through a nozzle with an orifice of 0.2 mm in diameter and is surrounded by a sheath-air flow which prevents the particle beam from diverging. The sampling flow is measured with a traceably-calibrated mass flow controller and is kept fixed at 60 mL/min, i.e. at a much lower flow rate than that of the bioaerosol monitors tested.

This ensures that coincidence losses, occurring when two or more particles cross the laser beam simultaneously, remain negligible. The laser beam is generated by a continuous-wave laser (Verdi V-5, Coherent, USA) at a wavelength of 532 nm, and is focused at the point of intersection with the aerosol stream using a cylindrical lens. Particles crossing the laser light sheet scatter light, which is then detected by a photomultiplier tube placed at 90 ° to the laser beam.

Typical particle number concentrations obtained with this setup range from $0.5~\mathrm{cm}^{-3}$ to several hundred $\mathrm{cm}^{-3}$ depending

on the PSL particle size (Horender et al., 2019). To achieve lower concentrations, a dilution unit can be placed upstream of the device under test (DUT). For example, to test the Swisens Poleno device at lower concentrations (see Subsection 3.1), the PSL number concentration in the aerosol homogeniser was set to about $1~\mathrm{cm}^{-3}$; the reference particle counter measured the





undiluted aerosol at $1\,\mathrm{cm}^{-3}$ to ensure enough particles were counted for statistical analysis while the Poleno device, which has a much higher sampling flow rate, sampled the diluted aerosol flow. Note that for the calculation of the reference concentration, $C_{\mathrm{ref}}$, the aerosol flow sampled by the reference OPC is always converted to volumetric flow at ambient temperature and pressure ($23 \pm 2\,^{\circ}\mathrm{C}$ and $960 \pm 10\,\mathrm{hPa}$).

## 2.2 Droplet Measurement Technologies WIBS-NEO

The WIBS-NEO (new version of the Wideband Integrated Bioaerosol Spectrometer) is developed and manufactured by Droplet Measurement Technologies (DMT). The device has a sheath flow rate of $2.1\,\mathrm{L/min}$ (compared with $2.4\,\mathrm{L/min}$ for its predecessor) of which it samples $0.3\,\mathrm{L/min}$ and was designed to identify particles in the size range from 0.5-30 μm in diameter. The WIBS-NEO uses two xenon lamps at 280 nm and 370 nm to excite fluorescence and two channels to detect the corresponding signals (310-400 nm and 420-650 nm). Furthermore, a 635 nm continuous laser is used for particle counting and sizing. If particle concentration, as detected by the sizing laser, is too high, the dead-time of the xenon lamps can introduce a duty cycle substantially below 100 % in the fluorescence measurement. Few studies exist using the WIBS-NEO given its novelty (Hughes et al., 2020), although (Forde et al., 2019) recently compared it with the performance of its predecessor, the WIBS-4, with which considerably more studies have been carried out (O'Connor et al., 2014; Perring et al., 2015b; Calvo et al., 2018).

## 2.3 Plair Rapid-E

The Plair Rapid-E samples $2.8\,\mathrm{L/min}$ of air and uses a 400 nm laser to acquire a scattered light signal at 24 angles ranging from 45-135 °. Each particle then interacts twice with a single pulse of a 337 nm laser and the resultant fluorescence is recorded in 32 spectral channels, each at a resolution of 14.51 nm per pixel within a spectral range of 350–800 nm and with eight sequential acquisitions with a retention of 500 ns. This rapid acquisition allows an estimation of the fluorescence lifetime in four spectral bands (350-400, 420-460, 511-572, 672-800 nm) and with a temporal resolution of 2 ns (Šauliene et al., 2019; Chappuis et al., 2019; Crouzy et al., 2016; Kiselev et al., 2013). For this experiment, the Rapid-E was operated in a mode for particles between 0.5 μm and 100 μm in diameter. In case of excessive noise on the detector, the thresholds are automatically adapted by the instrument, leading to a decreased sensitivity to small particles. Besides its research use, the Rapid-E has been employed operationally for airborne pollen monitoring as part of the RealForAll project (Tešendić et al., 2020).

### 2.3.1 Rapid-E particle size determination and spectrum correction

Particle optical diameter was calculated from the intensity of the scattering signal following the relationship provided by the manufacturer and described by algorithm 1 in appendix A. Essentially, the optical diameter is assumed to be proportional to the overall intensity of scattering signal by summing the detection over all angles for each particle measured.

The fluorescence spectrum from each particle was corrected before the analysis following the manufacturers guidelines using algorithm 2 in appendix A.



## 2.4 Swisens Poleno

The Swisens Poleno device makes use of two digital holography cameras to reconstruct an in-focus image of each particle.
Thereafter, a LED is employed to excite fluorescence at 280 nm and at 365 nm, after which the resultant fluorescence is recorded in five channels (335-380 nm, 415-455 nm, 465-500 nm, 540-580 nm, and 660-690 nm). A pump ensures air flows through the instrument at a rate of 40 L/min, while a virtual impactor unit (consisting of multiple virtual impactors in series) serves to concentrate particles with diameter larger than 5 µm (Sauvageat et al., 2020). According to the manufacturer the maximum enhancement factor of the virtual impactor unit is expected to be 1000 (personal communication, Swisens, May
2021). This improves counting statistics for large particles which are found at low number concentrations (typically $< 5\,\mathrm{cm}^{-3}$) in ambient air. At the same time, the virtual impactor filters out small particles (<1 µm), thus reducing the risk of coincidence losses. The quality of the images allows a first-level visual verification of particle classification, for example, between pollen and non-pollen, or even in certain cases to identify easily-distinguishable pollen taxa. The Poleno has to date been used only for research purposes, with up to eight different pollen taxa being identified (Sauvageat et al., 2020), although it is currently
being implemented across the Swiss operational pollen monitoring network.

### 2.4.1 Poleno particle size determination

The manufacturer does not provide a standardized method to determine particle size. We applied two methods to determine particle diameter. The first is based on the holographic images, from which the particle area is calculated by counting all pixels that make up the particle. The average area from both images is then used to compute the diameter assuming the
particles are spherical. From there, the diameter is converted from pixels to µm using the resolution of the holographic images (0.6 µm/pixel). This method is applicable only for particles with diameter of 10 µm and larger; smaller particles cannot be measured with this method because of limitations in the reconstruction software used at the time. For small particles, a second method, using the forward scattering signal from the trigger laser, is applied. This is done in three steps: the part of the raw signal up to a value of 800 is kept and then a high pass filter is applied to remove the base of the signal. A cut-off at a value
of 7880 is used, which is roughly equivalent to the median of the first part of the signal for 1 µm particles. Thereafter, the integral of the remaining signal is computed and the equation $d = aI^b$ using the median of the scattered light integral ($I$) for each particle diameter ($d$) is fitted, where $a = \frac{1}{630}$, and $b = 0.61$ (with $a$ and $b$ having been determined empirically from the presented data).

## 2.5 Experimental setup

Eight different types of PSL particles were used to test a range of sizes and fluorescence. Non-fluorescent particles from 0.5 µm to 10 µm were tested as well as red, blue, and plum purple fluorescent particles (see Table 1 for further details). The fluorescent particles were chosen to correspond to the detection wavelengths of each of the systems tested. PSL particles with diameters up to 2 µm were generated by nebulising suspensions while larger particles (i.e. 5 µm and 10 µm) were generated





from dry powder using a fluidised bed generator. Although necessary for a comprehensive calibration of bioaersol monitors,
no standardized method currently exists to produce accurate concentrations of PSL particles of diameter > 10 µm.

**Table 1.** PSL sphere types and nominal diameters, $d_{\mathrm{nom}}$, used in this study.

| PSL type | $d_{\mathrm{nom}}(\mu\mathrm{m})$ | Provider | Cat. No. |
|---|---|---|---|
| non-fluorescent | 0.5 | Thermo Scientific | 3500A |
| non-fluorescent | 0.9 | Duke Scientific Corporation | 1900A |
| non-fluorescent | 1.0 | JSR Corporation | - |
| plum purple | 2.07 | Bangs Laboratories, Inc. | FSPP005 |
| fluorescent red | 2.0 | Thermo Scientific | R0200 |
| fluorescent blue | 2.1 | Thermo Scientific | B0200 |
| non-fluorescent | 5.0 | Microbeads AS | CS5 |
| non-fluorescent | 10.0 | Microbeads AS | CA10 |

According to the PSL manufacturers, the refractive index of the non-fluorescent and the red/blue
fluorescent particles was 1.59 (589 nm, 25 °C). No information was available regarding the
refractive index of the plum purple PSL particles.

The number concentration measurements were carried out by connecting the bioaerosol devices one by one to the sampling
outlet of the primary standard described in Subsection 2.1. The nominal particle concentration was set to 0.5-2 $\mathrm{cm}^{-3}$ with
the bioaerosol monitors sampling the air flow at the same time as the reference optical particle counter. The latter yielded
the reference particle number concentration, $C_{\mathrm{ref}}$, with relative expanded uncertainties of 6-7% (coverage factor $k = 2$; 95%
confidence level).

For the measurements with the Rapid-E and WIBS-NEO no dilution of the test PSL aerosol was required. However, for
the Poleno, which uses a virtual impactor designed to concentrate particles larger than approximately 5 µm, the device has
an upper concentration threshold of about 300'000 $\mathrm{m}^{-3}$ (0.3 $\mathrm{cm}^{-3}$), requiring the use of a dilution system. The commercial
dilution unit HDS 561 (Topas GmbH, Germany), which can produce high dilution rates and is suitable for particle counters
with high volume flow, was placed upstream of the Poleno when sampling the 10 µm PSL particles.

## 3   Results and Discussion

### 3.1   Particle Number Concentration

#### 3.1.1   Counting Efficiency

The counting efficiency, $E_{\mathrm{DUT}}$, is calculated as $\sum_i E_{\mathrm{i,DUT}}/N$ with $E_{\mathrm{i,DUT}} = C_{\mathrm{i,DUT}}/C_{\mathrm{i,ref}}$, where $C_{\mathrm{i,DUT}}$ and $C_{\mathrm{i,ref}}$ are the
number concentrations measured by the DUT and the reference optical particle counter, respectively, and $N$ being the total
number of measurements (typically 25). Because of the higher flow rates and therefore higher particle counts registered by
the DUT compared to the reference counter, the statistical uncertainties of $C_{\mathrm{DUT}}$ are expected to be small (<10%). It must





be noted, however, that bioaerosol monitors can also suffer from large systematic errors, for instance, due to uncertainties in the aerosol flow rate, coincidence losses, impaction losses in the sampling system, misalignment of the laser beam, and size-

dependent particle losses in the concentrator. These systematic errors are in practice much larger than the statistical uncertainty of $C_{\mathrm{DUT}}$ or the uncertainty of $C_{\mathrm{ref}}$ (6-7%).

**Table 2.** Average particle concentrations measured by the three devices tested, $C_{\mathrm{DUT}}$, the reference optical particle counter, $C_{\mathrm{ref}}$, and counting efficiency of the devices tested, $E_{\mathrm{DUT}}$. $U$ denotes the expanded uncertainty (95% confidence level) of the counting efficiency.

| DUT | Poleno | | | Rapid-E | | | WIBS-NEO | | |
|---|---|---|---|---|---|---|---|---|---|
| $d_{\mathrm{nom}}(\mu m)$ | $C_{\mathrm{ref}}(\mathrm{cm}^{-3})$ | $C_{\mathrm{DUT}}(\mathrm{cm}^{-3})$ | $E_{\mathrm{DUT}} \pm U$ | $C_{\mathrm{ref}}(\mathrm{cm}^{-3})$ | $C_{\mathrm{DUT}}(\mathrm{cm}^{-3})$ | $E_{\mathrm{DUT}} \pm U$ | $C_{\mathrm{ref}}(\mathrm{cm}^{-3})$ | $C_{\mathrm{DUT}}(\mathrm{cm}^{-3})$ | $E_{\mathrm{DUT}} \pm U$ |
| 0.5 | 0.87 | $\approx 0$[1] | $\approx 0$ | 0.76 | $\approx 0$[1] | $\approx 0$ | 0.92 | 0.51 | $0.54 \pm 0.03$ |
| 0.9 | _[2] | _[2] | _[2] | _[2] | _[2] | _[2] | 1.27 | 1.13 | $0.90 \pm 0.06$ |
| 1.0 | 0.88 | $\approx 3.32$[1] | $\approx 3.81 \pm 0.47$ | 0.97 | $\approx 0$[1] | $\approx 0$ | _[2] | _[2] | _[2] |
| 2.07 | 1.14 | 0.93 | $0.82 \pm 0.14$ | 1.14 | $\approx 0.03$[1] | $\approx 0.02$ | 1.33 | 1.09 | $0.82 \pm 0.12$[5] |
| 2.0 | 1.54 | 1.50 | $0.98 \pm 0.13$ | 1.09 | $\approx 0.05$[1] | $\approx 0.04$ | 1.79 | 1.11 | $0.61 \pm 0.09$[5] |
| 2.1 | 1.02 | 0.95 | $0.95 \pm 0.18$ | 0.70 | $\approx 0.005$[1] | $\approx 0.01$ | 1.26 | 1.03 | $0.83 \pm 0.12$[5] |
| 5.0 | 2.19 | $17.43$[3] | $7.85 \pm 0.38$[3] | 2.39 | 0.98 | $0.42 \pm 0.04$ | 1.50 | 0.86 | $0.57 \pm 0.08$ |
| 10.0 | 1.87 | $214.65$[3,4] | $123.79 \pm 15.53$[3] | 1.69 | 0.94 | $0.58 \pm 0.06$ | 0.99 | 0.04 | $\approx 0.04$ |

[1]    The particle size is below or at the detection limit of the instrument.

[2]    The measurement sequence was defective or not performed.

[3]    Particle concentrator enhances the concentration of particles larger than 5 μm. Thus counting efficiency values larger than 1 are possible. The enhancement factor is particle size dependent and has a theoretical maximum of 1000.

[4]    Measurements were performed using a dilutor with a dilution factor of 55:1 (see section 2.5 for further details.)

[5]    The suspensions of the fluorescent PSL particles contained a considerable amount of additives, giving rise to a large peak of residue particles, which slightly overlapped with the peak of the PSL particles. It is also possible that the PSL particles have a thin organic coating due to the additives. In that case, the PSL diameter after nebulisation of the PSL suspension will be slightly larger than the geometric diameter reported by the manufacturers. The measurement uncertainties have been increased to account for a possible bias in the determination of the counting efficiency.

For small particles, the counting efficiency of the WIBS-NEO increases with particle diameter from 54 % for 0.5 μm to a maximum of 90 % for particles with 0.9 μm diameter (Table 2). This is in line with the results of Healy et al. (2012), who found counting efficiencies of 50% and 100 % for particles 489 nm and 690 nm in diameter, respectively. Beyond the peak at 0.9 μm,

the counting efficiency tends to drop with increasing particle diameters (61-83 % at 2 μm and 57 % for 5 μm particles, Table 2). The WIBS-NEO appears to reach an upper detection limit for particles larger than 10 μm, detecting just 4 % of the particles with a diameter of 10 μm compared to the reference counter. This observed drop in the detection rate for bigger particles is somewhat surprising since the WIBS-NEO is expected to be able to detect particles up to 30 μm.

        The Rapid-E shows a maximum counting efficiency of 58 % for the 10 μm PSL, while at 5 μm diameter 42 % of the particles

are detected. The lower detection limit was found to be around 2 μm, where only 1 to 4 % of the particles are detected. At 1 μm diameter, almost no PSL are detected by the Rapid-E (Table 2).

        The Poleno does not detect particles with 0.5 μm diameter and appears to have its lower detection limit at approximately 1 μm, where the counting efficiency is 3.81 (Table 2). This increased counting efficiency is potentially because the trigger laser was set too sensitively, resulting in noise being counted as particles during the 1 μm measurement sequence. At 2 μm the Poleno

performs well, with a counting efficiency ranging from 82-98 %. However an effect of the concentrator on the measurements





for particles at 2 µm diameter cannot be ruled out. Note that the counting efficiency of the Poleno strongly depends on the performance of the virtual impactor, also at small particle sizes. Small shifts in the cut-off curve of the impactor can have a large effect on the counting efficiency, for example, at 2 µm. For larger particles, the concentration measured by the Poleno is strongly enhanced due to the concentrator unit. It is expected that the plateau values of the concentrator's enrichment curve

are quite robust and could be used to correct concentration values with reasonably small uncertainty. However, the plateau efficiency could not be determined with the tested PSL diameters. For particles with 5 µm diameter, a concentration factor of 7.85, and for the particles with 10 µm diameter a factor of 123.79 was observed, which is still far below the theoretical maximal enhancement factor of 1000. This indicates that both tested diameters (5 and 10 µm) are on the lower end of the cut-off curve (Figure 1b). It is therefore not recommended to use the values in Table 2 to correct the concentration of the Poleno

measurements within this size range.

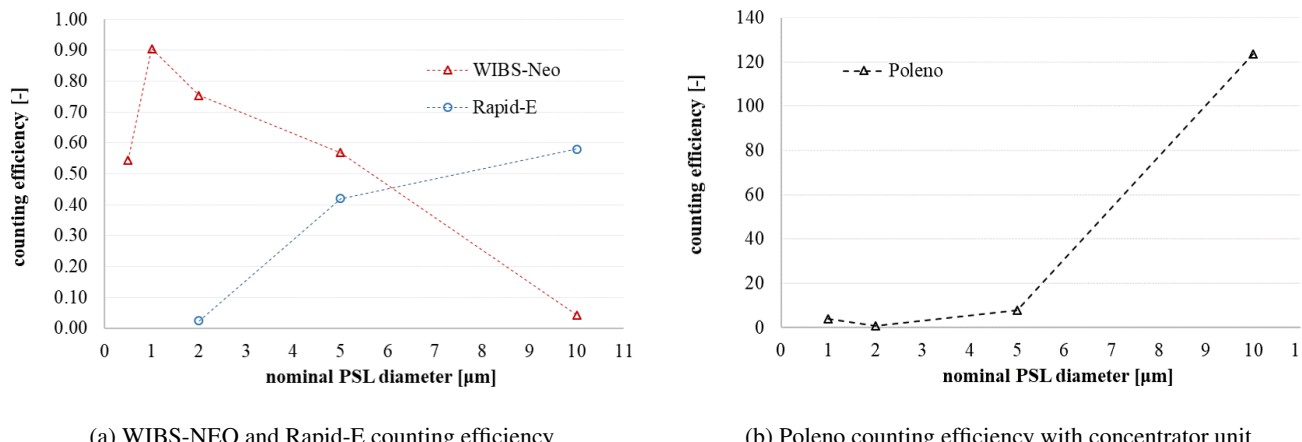

(a) WIBS-NEO and Rapid-E counting efficiency      (b) Poleno counting efficiency with concentrator unit

**Figure 1.** Counting efficiency for the three devices at different particle sizes. The value at 2 µm diameter is the mean value of the three measurement sequences at 2.0 µm, 2.07 µm, and 2.1 µm (i.e. red, plum purple, and blue, respectively).

### 3.1.2 Uncertainty and stability

A counting efficiency that is too low or high does not mean that a particular device is unable to provide valuable measurements in practice. If the counting efficiency is well known and stable, a scaling curve over the relevant size range can be determined for each instrument. To evaluate whether measurements are indeed scalable, we compute the root mean square error deviation

(RMSD) between the scaled reference and DUT counts. The RMSD is a combination of the uncertainties from the reference and DUT measurements as well as from other potential uncertainties related to the experimental set up. Since we cannot precisely assign the error to each of the potential sources we assume that the DUT is the only source of error and thus we can interpret the RMSD as an upper-bound of the error for each instrument. The lower the RMSD the better the measurements are suited for scaling without significantly increasing the uncertainty. The RMSDs for the different measurement sequences with

all particle diameters are given in table 3.



**Table 3.** Root mean squared deviation ($RMSD$) between the scaled DUT and the scaled reference concentration measurements. Both time series where scaled by dividing with their own average before computing the RMSD.

| | $RMSD(\%)$ | | |
|---|---|---|---|
| $d_{\mathrm{nom}}(\mu m)$ | Poleno | Rapid-E | WIBS-NEO |
| 0.5 | $-^{1}$ | $-^{1}$ | 0.11 |
| 0.9 | $-^{1}$ | $-^{1}$ | 0.10 |
| 1.0 | $0.27^{1}$ | $0.89^{1}$ | $-$ |
| 2.07 | 0.36 | 0.29 | 0.14 |
| 2.0 | 0.30 | 0.26 | 0.13 |
| 2.1 | 0.39 | 0.25 | 0.11 |
| 5.0 | 0.12 | 0.26 | 0.25 |
| 10.0 | $0.20^{2}$ | 0.29 | 0.25 |

[1] The particle size is below or at the detection limit of the instrument

[2] Measurements were performed using a dilutor with a dilution factor of 55:1 (see section 2.5 for further details.)

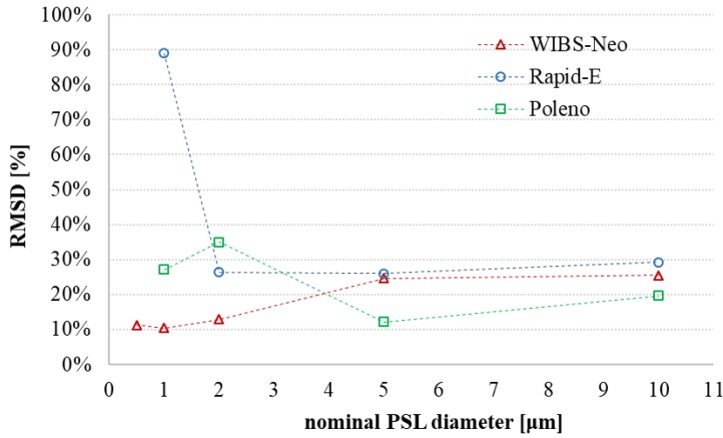

**Figure 2.** Root mean square deviation between normalized DUT and reference measurements. The value for 2 μm is the mean value of the three measurement sequences with similar diameter (2.07 μm, 2 μm, and 2.1 μm).

Out of the three devices tested, the WIBS-NEO is the only one able to perform robust measurements for particles with diameters from 0.5 μm to 2 μm (RMSD between 10% and 14%). For particles with larger diameters, the RMSD increases to 25% for the 5 and 10 μm PSL (Figure 2). The Rapid-E shows an RMSD of 25-29% for particles with diameters between 2 μm and 10 μm. For the 1 μm particles the RMSD of 89% indicates that the measurements are noisy and thus below the detection limit of the device. The Poleno shows relatively high RMSD values of 30-39% for the 2 μm particles, but a lower RMSD of 12% for the 5 μm particles. At 10 μm, the RMSD is slightly higher at 20%.





Note that all values are based on relatively small statistical samples ($\leq 30$ measurement points). It also has to be mentioned that the scaling factor can be very sensitive to particle size. In such a case, the capability of determining the particle size becomes an important uncertainty factor when scaling the concentrations.

## 3.2 Particle size determination

Each of the devices tested produce a scattering signal which can be used to estimate the particle size. While the devices are not specifically designed to do so, the measurements provide an indication of how accurate the optical detection of particles is. If the size distribution, measured for a specific PSL size, shows a well defined peak this gives an indication that the DUT is able to detect particles of that size accurately.

### 3.2.1 Droplet Measurement Technologies WIBS-NEO

(a) 0.5 μm

(b) 1 μm

(c) 2 μm

(d) 5 μm and 10 μm

**Figure 3.** Size distribution estimated by the WIBS-NEO for different PSL particles with diameters ranging from 0.5-10 μm





The WIBS-NEO has well distinguished peaks at 0.9 µm, 1 µm, 5 µm, and 10 µm (Figure 3b and 3d) and thus appears to provide precise sizing for each particle within this size range. Deviations of (geometric) mean optical diameter measured for a certain PSL sample from nominal geometric diameter are small and within uncertainty of optical sizing methods. The measurement at 0.5 µm shown in Figure 3a may be truncated at the small size end due to the lower size detection limit of the WIBS (see also counting efficiency results in Section 3.1.1). Interestingly, the different types of PSL particles tested appear to impact the measurements to a certain extent. In particular, for the 2 µm particles, the red PSL show a peak at 3.3 µm while the plum purple and blue PSL peaks are closer to the actual particle optical diameters of 2.3 µm and 2.2 µm, respectively (Figure 3c). The variation in measured (geometric) mean diameters for the 3 different PSL samples with nominal geometric diameters between 2.0 to 2.1 µm is considerably greater than expected optical sizing precision. In particular the red PSLs appear at around double the nominal size. This suggests a larger scattering cross-section of the red PSLs compared to the blue and plume purple PSLs at the wavelength of the sizing laser (635 nm).

### 3.2.2 Plair Rapid-E

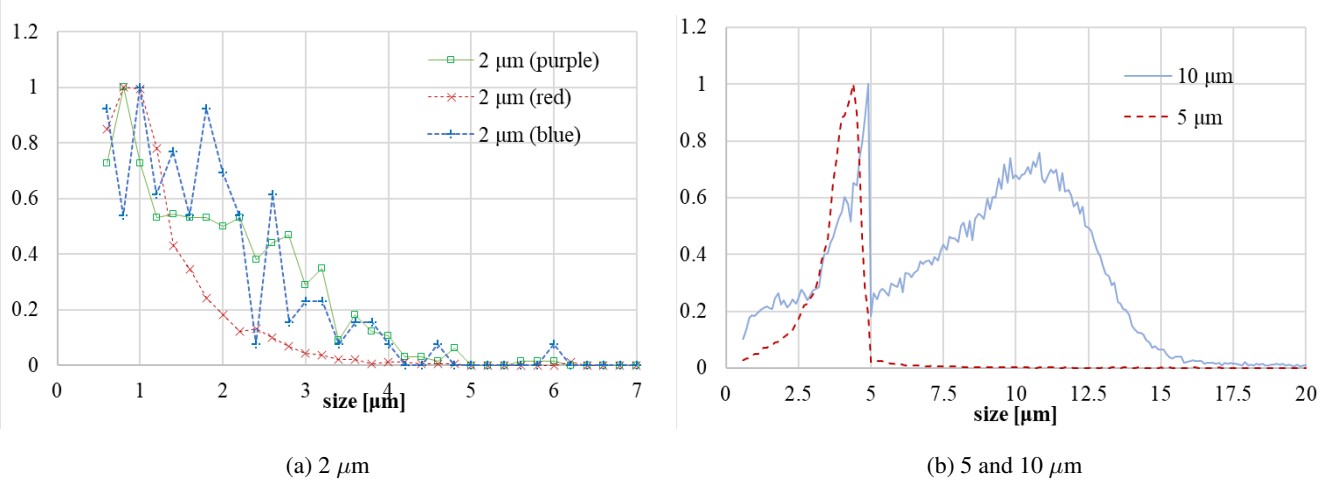

(a) 2 $\mu$m                                    (b) 5 and 10 $\mu$m

**Figure 4.** Size distribution estimated by the Plair Rapid-E for different PSL particles with diameters ranging from 2-10 µm.

The Rapid-E was unable to detect 1 µm particles and does not show a distinct peak for any of the 2 µm PSLs either. For these particles the size distribution curve shows no clean peak (figure 4a), indicating that the detection limit is likely above 2 µm for this device. The peak for the 5 µm particles is well-defined and therefore indicates that the Rapid-E is able to accurately detect particles in this size. For the size distribution of the the 10 µm diameter particles, a first narrow peak just below 5 µm, and a second broader peak at 10 µm with a large tail towards smaller diameters can be observed in figure 4b. The large tail towards lower sizes could be an indication of a misalignment of the laser. The discontinuity of the size distributions at 5 µm in figure 4b is a direct consequence of using an intensity-to-particle relationship with a discontinuous first derivative (figure 5, Algorithm



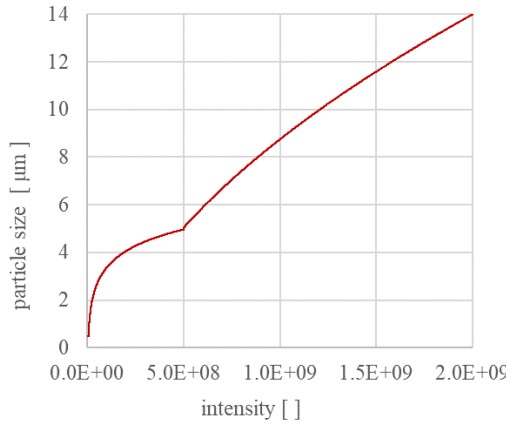

**Figure 5.** The relationship between light scattering intensity and particle size for the Plair Rapid-E as recommended by the manufacturer. Note that two different equations are used to calculate size for particles smaller and larger than 5 μm in diameter, respectively.

1 in appendix A). This suggest potential for improving the sizing method for particles with diameters smaller than 10 um, but this was not the scope of this study.

### 3.2.3 Swisens Poleno

The Swisens Poleno is unable to detect 0.5 μm diameter particles. For the 1 μm diameter PSL particles, the size distribution shows a well defined peak at 1.05 μm (Figure 6a), indicating that the device is able to distinguish particles in this size range well. At 2 μm diameter (Figure 6b), the size distributions show well-defined peaks for the red, plum-purple, and blue particles, with slight differences in positioning around 2 μm. The 5 μm size distributions shows a well defined peak, although the size is slightly overestimated (Figure 6c). The 10 μm particles size distribution also shows a well-defined peak, but again the size is overestimated at about 10.7 μm. The shoulder of the 5- and 10 μm size distributions at smaller particle sizes may indicate the need to realign the laser and particle beams and/or the detection optics. When comparing the two methods of particle size estimation used for the Poleno device (see section 2.4.1), the method based on the holographic images results in a very clean and sharp peak compared to the trigger signal method (Figure 6d). This suggests that the image based method should be used for the Poleno device for large particles.

### 3.3 Fluorescence Measurements

The fluorescent response after excitation at different wavelengths is measured in all the three devices using three different PSL particles (blue, plum purple, and red) with 2 μm diameter. These measurements were then compared to reference data from the Max Planck Institute for Chemistry (Könemann et al. (2018)). The intensity values obtained from the devices tested were scaled to the reference curve with high spectral resolution. The peaks due to first- and second-order elastic scattering were





(a) 1 μm

(b) 2 μm

(c) 5 μm

(d) 10 μm

**Figure 6.** Size distribution estimated by the Swisens Poleno for different PSL particles with diameters ranging from 1-10 μm. The estimates where computed based on the trigger laser signal, except for the '10 μm (IMG)' curve in fig. 6d where the image-based method was used.

removed from the reference data as well as from each device tested (e.g. for excitation at $\lambda = 365$ nm, the peaks at $\lambda = 365$ nm and at $2\lambda = 730$ nm were removed).

### 3.3.1 Droplet Measurement Technologies WIBS-NEO

The WIBS device provides a maximum of two data points per excitation wavelength, therefore no robust and meaningful scaling to the available reference data was possible, and no further analysis was performed for this device.

### 3.3.2 Plair Rapid-E

The Rapid-E's fluorescence spectra correspond relatively well with those from the reference data (Figure 7). For the plum purple PSL particles ($\lambda = 425$ nm) both the reference and Rapid-E exhibit a broad peak in the range $\lambda = 400$-450 nm, with





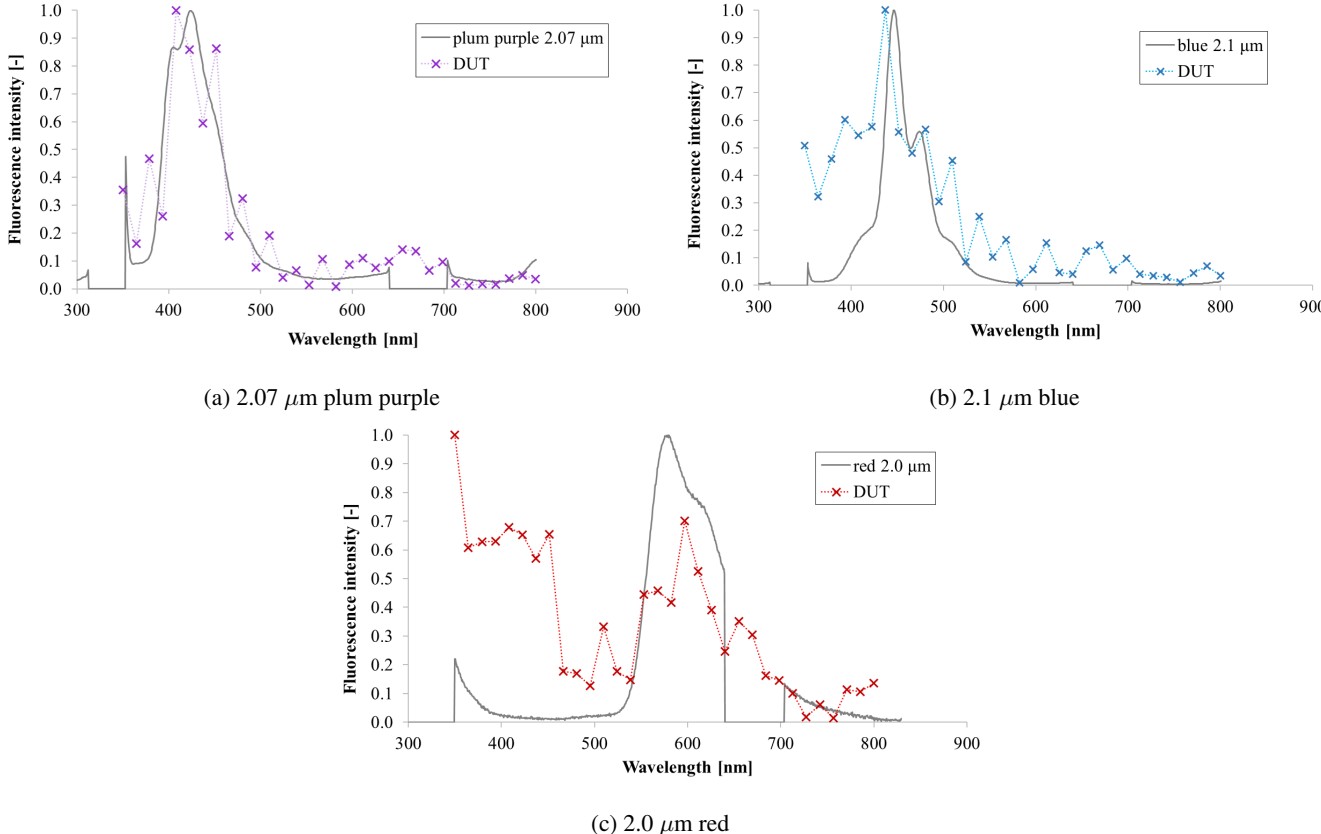

(a) 2.07 $\mu$m plum purple

(b) 2.1 $\mu$m blue

(c) 2.0 $\mu$m red

**Figure 7.** Comparison of the Rapid-E fluorescence measurements (DUT) with the reference measurements from Könemann et al. (2019) for: (a) Plum purple PSL particles with expected peak at 422 nm, (b) blue PSL particles with expected peak at 447 nm, and (c) red PSL particles with expected peak at 582 nm. All measurements are scaled to a maximum value of 1.

the Rapid-E slightly shifted towards longer wavelengths. For the blue PSL particles ($\lambda = 447$ nm) the opposite is true, with the Rapid-E signal slightly shifted towards shorter wavelengths and with a significant tail in this direction. The red PSL particles ($\lambda = 579$ nm) show a less defined peak in the expected range (around 580 nm but there is a strong signal from 400-450 nm, where no peak is seen in the reference data. Table 4 shows the percentage of particles measured by the Rapid-E with peak
fluorescence emission at the correct wavelength. Despite the spectra not showing peaks at the expected wavelengths, a notable number of measured particles show maximum fluorescence at these wavelengths.

### 3.3.3 Swisens Poleno

The reference data, when averaged for the different Poleno windows, are very similar for all excitation wavelengths once the first- and second-order elastic scattering peaks have been removed (figure 8). Overall, the Poleno compares well with the
reference data for the three different PSL tested. The limited spectral resolution of the device does not allow the same level of





**Table 4.** Number and percentage of particles with "almost correct" fluorescence peak position measured by the Rapid-E. In the third column two cases are distinguished: ±10 nm and ±20 nm deviation from the reference peak wavelength.

| Group | Total Particles | Reference Peak [nm] | Meas. Peak [nm] | | Particles with Correct Peak | |
|---|---|---|---|---|---|---|
| | | | Tolerance | Selected Peak | Number | Percentage |
| Plum Purple | 1571 | 425 | ±10 | 422 | 299 | 19.0 |
| | | | ±20 | 408,422,437 | 761 | 48.44 |
| Red | 2766 | 579 | ±10 | 582 | 110 | 3.98 |
| | | | ±20 | 567, 582, 596 | 407 | 14.71 |
| Blue | 274 | 447 | ±10 | 437, 451 | 86 | 30.66 |
| | | | ±20 | 437, 451, 466 | 102 | 36.96 |

comparison as the Rapid-E, however, the available measurement points are all close to the reference data. Note that the values for the 335-380 nm detection window are discarded as a result of the first-order elastic scattering so no points are shown for this wavelength range.

## 4   Conclusions

Three commercially-available bioaerosol monitors were tested for their ability to detect and count particles at different sizes and for the accuracy of their fluorescence measurements. A range of different polystyrene (PSL) particles was tested in a state-of-the-art calibration facility at the Federal Institute for Metrology METAS.

The DMT WIBS-NEO showed very good counting efficiency for particles with diameters of 1 μm, however, as particle size increases its detection performance was shown to drop considerably. This result was somewhat unexpected and this issue

remains to be investigated in more depth. The Plair Rapid-E was shown to be better suited for detecting larger particles, with detection performance increasing with particle size. The counting efficiency of the Poleno detection system could not be accurately estimated since the integrated virtual impactor unit was found to work at the lower end of the cut-off curve for the tested particle sizes. In order to determine the plateau efficiency of the virtual impactor unit, further measurements with larger particles sizes are necessary.

The RMSD analysis of the different measurement sequences confirms that the WIBS-NEO performs best for smaller particles (<= 2 μm). Only the measurement sequence from the Poleno at 5 μm and possibly at 10 μm showed similarly low RMSD values (10-20%). The particle size determination of the WIBS-Neo agreed with the reference, with well-shaped particle size distribution curves for all tested particles. However, for the fluorescent 2 μm particles, the red particles were estimated to be considerably larger than the blue and plum purple particles. Whether this is an effect of a stronger scattering response due to

the particle colour or an effect of the particle being coated in additives and becoming physically bigger is speculative at this point and should be investigated further. For the Rapid-E device, the algorithm for determining particle size delivered by the manufacturer led to some artifacts in the size distribution curve. This method could be improved to obtain more robust results







(a) 2.07 $\mu$m plum purple

(b) 2.1 $\mu$m blue

(c) 2.0 $\mu$m red

**Figure 8.** Comparison of the reference (REF) fluorescence measurements from Könemann et al. (2019) to the measurements with the Poleno instrument (DUT). All measurements are scaled to a maximum value of 1.

since the measurements are likely to be sufficient to do so. Finally, two methods of size determination were tested for the Poleno device. It was shown that the image-based approach works better than the scattering signal one, however, it can only be applied to larger particles.

Results show that the detection limit of the WIBS-NEO is below 0.5 μm diameter, the smallest particle size tested here. In contrast, the minimum detection limits for the other two instruments appear to be between 0.5 μm and 1 μm in diameter for the Poleno, and between 2 μm and 5 μm diameter for the Rapid-E.

The Rapid-E fluorescence measurements compare well with the reference data, even tough the signal to noise ration seem to be elevated for the red PSL particles. The Poleno device aligns well with the reference data and shows a very good correlation. Nevertheless, the coarse spectral resolution of the Poleno does not allow as precise a comparison as for the Rapid-E. A vali-



dation of the WIBS-NEO fluorescence observations was not possible because of the very low spectral resolution of the device (see section 3.3.1). In future the stability of the fluorescence measurements in time and amongst different devices of the same type should be tested.

Our results show that the three devices tested all have different strengths and weaknesses which need to be taken into account when considering an instrument for a specific application. In terms of pollen monitoring, it would be relevant to test even larger particles (e.g. 20 µm) particularly since the Rapid-E and Poleno were developed to detect airborne pollen which typically have diameters ranging from 10-100 µm in size. At present, no methods exist to produce accurate concentrations of PSL particles of this size, nor, of even greater interest, of fresh pollen particles. These challenges remain to be resolved to allow standardized

calibration of bioaerosol monitors for their target particle size, and thus to more fully understand what is present under real atmospheric conditions.

*Code and data availability.* Data and algorithms presented in this paper are experimental and subject to further development. They are available for research purposes on request to the authors of the paper.

*Author contributions.* KA, BCr, FT, and KV designed and carried out the experiments. KA, KV, JM, DO, BS, AM, GL, BCr, and FT

contributed to the data analysis. All authors contributed to the interpretation of the results and to writing this paper.

*Competing interests.* The authors declare that they have no conflict of interest.

*Acknowledgements.* This work is a contribution to the EUMETNET AutoPollen Programme. Part of this work was carried out within the Aeromet 2 project which has received funding from the EMPIR programme. The EMPIR programme is co-financed by the Participating States and from the European Union's Horizon 2020 research and Innovation Programme. Branko Sikoparija acknowledges financial support

from the Ministry of Education, Science and Technological Development of the Republic of Serbia (Grant No. 451-03-9/2021-14/200358) and the BREATHE project funded by the Science Fund of the Republic of Serbia (Grant No. 6039613).

**Appendix A:  Algorithms**



---

**Algorithm 1** Plair Rapid-E particle size determination

---

**Input**: scattering image $M \in \mathbb{R}^{24 \times N}$, N-number of acquisitions

**Output**: particle size $p \in \mathbb{R}$

1: $x = sum(M_{ij}, i \leftarrow 1 : 24, j \leftarrow 1 : N)$

2: **if** $x < 5500000$ **then**

3:     $p = 0.5$

4: **else if** $(x \geq 5500000)$ **and** $(x < 500000000)$ **then**

5:     $p = 9.95e - 1 \cdot log(3.81e - 5 \cdot x) - 4.84$

6: **else**

7:     $p = 0.0004 \cdot x^{0.5} - 3.9$

8: **end if**

---

**Algorithm 2** Plair Rapid-E fluorescence spectrum correction

---

**Input**: fluorescence spectrum $S \in \mathbb{R}^{32 \times 8}, S = [s_{ki}]_{32 \times 8}$, where $s_i \in \mathbb{R}^{32 \times 1}$ is $i$-th acquisition

**Output**: $c = [c_1, c_2, ..., c_{32}] \in \mathbb{R}^{32 \times 1}$

1: $m = min(s_{k7}, s_{k8}, \text{ for } k \leftarrow 1 : 32)$

2: **if** $any(s_{k1} > 20000, \text{ for } k \leftarrow 1 : 32)$ **then**

3:     $c \leftarrow s_2 + s_3 + s_4 + s_5 - 4 \cdot m$

4: **else**

5:     $c \leftarrow s_1 + s_2 + s_3 + s_4 - 4 \cdot m$

6: **end if**

7: **if** $max(c_k, \text{for } k \leftarrow 1 : 32) > 0$ **then**

8:     **for** $k \leftarrow 1 : 32$ **do**

9:         $c_k \leftarrow max(0, c_k)$

10:         $c_k \leftarrow c_k / max(c_k, \text{for } k \leftarrow 1 : 32)$

11:     **end for**

12: **else**

13:     $c \leftarrow [0, 0, ..., 0] \in \mathbb{R}^{32 \times 1}$

14: **end if**

---



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
