# Peer review of "Assessment of Real-time Bioaerosol Particle Counters using Reference Chamber Experiments"

_Atmospheric Measurement Techniques, 2021_

## Author Response (AR1)

**Review of "Assessment of Real-time Bioaerosol Particle Counters using Reference Chamber Experiments" by Lieberherr et al.**

**RC1**: 'Comment on amt-2021-136', Ghislain Motos, 30 Jun 2021
     **AC1**: 'Reply on RC1', Gian Lieberherr, 22 Jul 2021

*Reviewer: This paper presents interesting data in a very concise and understandable way. The limitations of the study are explained, and further improvements are detailed.*

**Response**: Thank you very much for the positive feedback.

*Reviewer: I noted very few minor comments:*

*Reviewer: Lines 88, 98, 114: Does AMT require to specify the address of the manufacturer (at least region and country)?*

**Response**: Thank you for pointing this out. We have now amended the text as follows: (DMT, Colorado, USA)

*Reviewer: Line 173: It is interesting to note that the WIBS-NEO was unable to detect 10 um PSL, where it should operate up to 30 um. It would be worth digging a bit more here to at least give hypotheses on what could have happened. What does literature say about detection efficiency of this model of the WIBS for particles larger than 5 um? If some studies showed opposite results than yours, where did something go wrong?*

**Response**: The results are indeed surprising. To the best of our knowledge, only one paper on the counting efficiency of the WIBS-4 by Healey et al. (JAS, 2012) exists. In this study, the authors calibrated the WIBS-4 against a CPC for particles in the size range 0.3 µm – 1.3 µm. We are not aware of any other study on the counting efficiency of the WIBS-4 or WIBS-NEO at larger particle sizes.

It is nevertheless unfortunate that we only calibrated a single WIBS-NEO in our study. Instrument performance depends on several factors, including proper maintenance. Ideally, a calibration campaign should include several devices of the same model to study unit-to-unit variability and correlate instrument performance with maintenance state and service years of the device. However, due to limited availability of WIBS instruments in Switzerland and at partner institutes abroad, we were only able to investigate only one WIBS-NEO in our study.

To avoid any misunderstandings, we now provide more information on the maintenance status of the instruments.

In Subsection 2.2: The WIBS-NEO employed in our study underwent a service at the manufacturer in early 2017.

In Subsection 2.3: The Rapid-E underwent a service in 2018.

In Subsection 2.4: The Poleno was used as received from the factory.

In Subsection 3.1.1 – Line 173, we amended the text as follows: "This observed drop in the detection rate for bigger particles is somewhat surprising since the WIBS-NEO is expected to be able to detect particles up to 30 µm. Although the exact reasons for this behaviour are unclear, it cannot be ruled out that the specific WIBS-NEO used in our study required a technical service by the manufacturer".

Section 4: More studies are needed to investigate the unit-to-unit variability of the Poleno, Rapid-E and WIBS-NEO bioaerosol monitors. It is known that inadequate maintenance can affect both the sizing accuracy and the counting efficiency of light-scattering instruments (Vasilatou, 2021).

References: K. Vasilatou et al. "Calibration of optical particle size spectrometers against a primary standard: Counting efficiency profile of the TSI Model 3330 OPS and Grimm 11-D monitor in the particle size range from 300 nm to 10 µm", *Journal of Aerosol Science* **157**, 105818 (2021).

*Reviewer: -Line 175: You mention a lower detection limit at around 2 um for the Rapid-E, but you only tested 2 um (no detection) and 5 um (98% detection efficiency). How do you know the lower detection limit is not 3 um, or 4 um?*

**Response**: We calibrated the Rapid-E monitor at particles sizes 0.5 µm, 1 µm, 2 µm, 5 µm and 10 µm. The detection efficiency is zero at 0.5 µm and 1 µm, but rises to 5% at 2 µm. These data indicate that the Rapid-E already starts to detect particles at 2 µm.

Please note that at 5 µm the detection efficiency is 42% (please see 7th column of Table 2).

*Reviewer: -Figure 2: No need to repeat "%"symbol in the y-axis labels, this is already in the legend.*

**Response:** If the Reviewer agrees, we would like to keep the % symbol in the y-axis for clarity.

*Reviewer: -Line 256: You mention the WIBS did not give fluorescence results because it "provides a maximum of two data points per excitation wavelength". This is difficult to understand for non-WIBS experts. Could you detail what prevent you from showing WIBS results here? This is a relatively big limitation of the paper, so this would deserve more explanation.*

**Response:**

We apologise that this section was not clear. We have changed the text to read as follows:

"Since the WIBS-NEO only detects signals in 2 channels (310-400nm and 420-650nm) a maximum of two data points per excitation wavelength are obtained. One of these points is used to scale to the reference values, so only one value remains to evaluate the signal. No meaningful analysis is thus possible and no further assessment was performed for this device for fluorescence."

**Response:** We thank you for your time and valuable feedback.

*Reviewer: General Comment*

*The present study performed a direct comparison of three different bioaerosol devices (i.e., the WIBS-NEO, Plair Rapid-E, and Swisens Poleno) to assess their performance in counting and sizing aerosol particles, as well as their accuracy of their fluorescence measurements. The authors highlighted the different strengths and weaknesses of the three devices for the evaluated particle size range. This is well designed study and the manuscript is well written. The results are valuable for the aerosol community and provides important information for future studies. The manuscript can be accepted after the following minor comments are properly addressed.*

**Response:** Thank you very much for the positive feedback

*Reviewer: Minor Comments:*

1. *I do not see the point to have Table 3 and Figure 2 as they show the same results.*

**Response:** It is correct that Figure 2 illustrates information from table 3. However, the value at 2 µm in Figure 2 is the average of the three 2-µm PSL values from table 3. This is also the case in Figure 1 illustrating the information from Table 2. We think that the figures help to understand the values from the tables. If the reviewer agrees, we suggest keeping the figures.

2. *Figures 3, 4 and 6. I am wondering how many experiments were performed to get each PSD.*

**Response:** The duration of the measurement sequences was about 20 minutes for each PSL and DUT combination (i.e. one experiment per size/fluorescence per instrument). To clarify this in the text we have added a sentence in section 2.5 at line 147:

"The number concentration measurements were carried out **for each PSL**, by connecting the DUTs  to the sampling outlet of the primary standard described in Subsection 2.1. **Once the concentration was stabilized, the measurement sequences lasted for 20 minutes.** The nominal particle concentration was set to …"

*3. Figures 7 and 8. I am wondering how many experiments were performed to get these figures.*

**Response:** See response to comment 2 above

*4. Lines 224-225: "appear at around double the nominal size". I consider that it is quite far from double. Can the authors be more precise here?*

**Response:** We completely agree, the sentence has been changed to: " In particular the red PSLs appear at around 1.6 times the nominal size."

*5. Lines 249: "is measured in all the three devices". I suggest to changes this to "two devices" and to add the text from section 3.3.1. I mean, I consider useless to have subsection 3.1.1.*

**Response:** We suggest to change in line 249: "…is **evaluated on the DUT's** using three different PSL…"

We also agree concerning subsection 3.3.1. We have removed the title of 3.1.1, and then directly added the content from subsection 3.3.1. to the end of the paragraph.

(Subsection 3.3.1 was reformulated within the response to RC1 : "Since the WIBS-NEO only detects signals in 2 channels (310-400nm and 420-650nm) a maximum of two data points per excitation wavelength are obtained. One of these points is used to scale to the reference values, so only one value remains to evaluate the signal. No meaningful analysis is thus possible and no further assessment was performed for this device for fluorescence.")

*6. Lines 250: "with 2 µm diameter". I am wondering why at this specific size only?*

**Response:** The choice of particles was limited to the ones used in the reference dataset (Könemann et al.). The 2um size had the advantage that all the suited colors where available in the same size. Furthermore, the overall number of experiments was limited as the measurements were time consuming (each measurement sequence had to be prepared carefully which added considerable overhead to the effective runtime).

*Reviewer: Technical Comments:*

*Reviewer: Line 17 and along the text: I suggest to organize the references chronologically.*

**Response:** Indeed, this will be adapted, thanks for the remark.

*Reviewer: Line 26: Define WIBS.*

**Response:** This has been added: ' WIBS (Wideband Integrated Bioaerosol Sensor)'

*Reviewer: Line 41: Add a reference after "methods".*

**Response:** We add the following reference:

 Löndahl J. (2014) Physical and Biological Properties of Bioaerosols. In: Jonsson P., Olofsson G., Tjärnhage T. (eds) Bioaerosol Detection Technologies. Integrated Analytical Systems. Springer, New York, NY. https://doi.org/10.1007/978-1-4419-5582-1_3

*Reviewer: Line 95: "(Forde et al., 2019)" should be Forde et al. (2019).*

**Response:** Thank you for the remark. This has been modified as suggested.

*Reviewer: Line 144: Add the model and manufacturer of the used fluidized bed generator.*

**Response:** Thank you for the remark. We have added a parenthesis to the end of the sentence: '…using a fluidized bed generator (3400A, TSI Inc., USA).'

*Reviewer: Line 146: Should "bioaerosol devices" be replaced by "DUTs".*

**Response:** Thank you for the remark. This has been modified as suggested.

*Reviewer: Line 148: Should "bioaerosol monitors" be replaced by "DUTs".*

**Response:** Thank you for the remark. This has been modified as suggested.

*Reviewer: Lines 200, 229 and along the text: Table(s) and Figure(s) should have a capital "T" and "F", respectively.*

**Response:** Thank you for the remark. This has been corrected.

*Reviewer: Lines 251: "(Könemann et al. (2018))" should be "(Könemann et al. 2018)".*

**Response:** Thank you for the remark. This has been modified as suggested.

*Reviewer: Figure 8: The panels are cut-off at the bottom.*

**Response:** Unfortunately, we do not see the figure 8 being cut-off. Would it be possible to explain in more detail what the issue appears to be? Thank you.

**Response:** We thank you for your time and valuable feedback.

---

## Author Response (AR2)

**Associate Editor decision: Publish subject to minor revisions (review by editor)**
by Zamin A. KanjiComments to the author

Lines and pages refer to revised manuscript with track changes (yellow highlights).

Abstract Line 12-14: The authors mention that the method suggested in this paper could be used as potential standardised validation method. Indeed if this is the case, I would strongly suggest the authors include a schematic how their experimental validation set up looked like. This would be important and would likely help future readers the community to adapt the method herein as a standardised validation method.
*Answer: An additional figure (fig.1) with the schematic was added to the manuscript.*

Based on the reviewer comments the authors have included the last service dates of the instruments used. However these additions bear no meaning if the dates of when the measurements were conducted are not give. I would suggest to include this, so there is a reference point as to how long before the measurements did the instruments receive their service. Suggestion to include the measurement date in Section 2.1
*Answer: We added that, but if the editor agrees we would like to add it into the introduction (line 45) instead of section 2.1 as suggested.*

Section 2.4.1 line 133: Her it was not clear to me when the authors state "The average area from both images is then ..." at this point I do not follow wha both refers to. Since you are still discussing only the first method. Can you please clarify.
*Answer: For the first method, both available holographic images were used. The text should now be clearer.*

Reviewer 2 asks for the number of measurements conducted. In page 6 line 152-153 you mention that the measurement sequence lasted for 20 minutes. But I see this as not answering the question. How many of the 20 minute sequences were conducted? Was it just one? In which case you should clarify on the sufficiency of just one measurement. i.e. if the instruments are single particle instruments, then in the 20 minutes you have likely sampled many particles, which would be one reason I can think of why not at least 3 or more of the 20 minute sequences were not conducted.
*Answer: I modified the text and I hope this is better explained now. There where continuous measurements during 20 minutes, and thus up to several hundreds of particles per minute where measured one by one. Guessing that this would not be the requested answer we tried to better explain how the measurements were performed. (We computed 1-minute concentrations, which results into 20 values to compare between DUT and reference OPC. I hope this is clear in the text)*

Reviewer 1 suggested removing the "%" sign from the y-axis of Fig. 2. I support this edit, however, if you want to keep the sign, then I suggest only labelling the y-axis every 20% (i.e. 0%, 20%, 40%) because the figure y-axis appears quite crowded in its current format.
*Answer: We implemented both of your suggestions: removing "%" and introducing larger step sizes (20%)*

Once the above revisions (or responses) are done, I would be happy to accept the publication.
*Answer: Thanks for your remarks and suggestions, we think that the manuscript gained in quality once more.*